# The *nodD1* Gene of *Sinorhizobium fredii* HH103 Restores Nodulation Capacity on Bean in a *Rhizobium tropici* CIAT 899 *nodD1*/*nodD2* Mutant, but the Secondary Symbiotic Regulators *nolR*, *nodD2* or *syrM* Prevent HH103 to Nodulate with This Legume

**DOI:** 10.3390/microorganisms10010139

**Published:** 2022-01-10

**Authors:** Francisco Fuentes-Romero, Pilar Navarro-Gómez, Paula Ayala-García, Isamar Moyano-Bravo, Francisco-Javier López-Baena, Francisco Pérez-Montaño, Francisco-Javier Ollero-Márquez, Sebastián Acosta-Jurado, José-María Vinardell

**Affiliations:** Department of Microbiology, University of Sevilla, Avda. Reina Mercedes 6, 41012 Seville, Spain; ffuentesr@us.es (F.F.-R.); pnavarro2@us.es (P.N.-G.); payala@us.es (P.A.-G.); isamar_mb_1991@hotmail.com (I.M.-B.); jlopez@us.es (F.-J.L.-B.); fperezm@us.es (F.P.-M.); fjom@us.es (F.-J.O.-M.)

**Keywords:** rhizobia–legume symbiosis, Nod factors, *Phaseolus vulgaris*, *Rhizobium tropici* CIAT 899, *Sinorhizobium fredii* HH103, *nodD1*, *nodD2*, *nolR*, *syrM*, *ttsI*

## Abstract

Rhizobial NodD proteins and appropriate flavonoids induce rhizobial nodulation gene expression. In this study, we show that the *nodD1* gene of *Sinorhizobium fredii* HH103, but not the *nodD2* gene, can restore the nodulation capacity of a double *nodD1*/*nodD2* mutant of *Rhizobium tropici* CIAT 899 in bean plants (*Phaseolus vulgaris*). *S. fredii* HH103 only induces pseudonodules in beans. We have also studied whether the mutation of different symbiotic regulatory genes may affect the symbiotic interaction of HH103 with beans: *ttsI* (the positive regulator of the symbiotic type 3 protein secretion system), and *nodD2*, *nolR* and *syrM* (all of them controlling the level of Nod factor production). Inactivation of either *nodD2*, *nolR* or *syrM*, but not that of *ttsI*, affected positively the symbiotic behavior of HH103 with beans, leading to the formation of colonized nodules. Acetylene reduction assays showed certain levels of nitrogenase activity that were higher in the case of the *nodD2* and *nolR* mutants. Similar results have been previously obtained by our group with the model legume *Lotus japonicus*. Hence, the results obtained in the present work confirm that repression of Nod factor production, provided by either NodD2, NolR or SyrM, prevents HH103 to effectively nodulate several putative host plants.

## 1. Introduction

Rhizobia are a diverse group of α- and β-proteobacteria able to enter in symbiosis with legumes [1]. In this relationship, rhizobia are able to infect legume roots and induce the formation of new organs called nodules. Eventually, rhizobia invade nodule cells by endocytosis and differentiate into bacteroids, which express the nitrogenase genes and, therefore, can fix N_2_ into ammonia [2]. Most of the ammonia obtained is given up to the plant that, in turns, feeds the bacteroids with carbon and nutrients. The whole process is commonly referred as nodulation. This symbiosis is extremely important since it can eliminate the necessity of the application of nitrogen fertilizers, which are expensive and highly polluting in agriculture practices [3,4]. That is why understanding the signaling mechanisms that operate in this symbiosis may help to improve the yield of legume crops, which could be extremely beneficial for sustainable agriculture [4].

The rhizobia–legume symbiosis is highly specific: each rhizobial strain establishes successful symbiotic interactions with specific host legumes [5]. *Rhizobium tropici* CIAT 899 and *Sinorhizobium fredii* HH103 are two of the best studied rhizobial strains so far [6,7]. *R. tropici* CIAT 899 can establish effective symbiosis with several legumes such as *Phaseolus vulgaris* (bean) and the model legume *Lotus japonicus* [8,9,10], but only induces pseudonodules unable to fix nitrogen on soybean plants [11]. *Sinorhizobium fredii* HH103 is a broad-host range rhizobial strain since it can nodulate several dozens of legumes, including the important crop *Glycine max* and *Lotus burttii*, but not *L. japonicus* or *Phaseolus vulgaris* [7,12]. Thus, CIAT 899 and HH103 are agronomical relevant rhizobia since they are symbionts of highly important crops: bean and soybean, respectively.

The rhizobia-legume symbiosis relies on a complex molecular dialogue, which explains, at least partially, the specificity of this interaction [2,13]. This dialogue starts with the root exudation of phenolic compounds, flavonoids, that interact with and activate NodD, which belongs to the LysR family of bacterial transcriptional regulators. The activated NodD protein binds to regulatory conserved sequences called *nod* boxes, located upstream of the bacterial nodulation genes, inducing its expression and, thus, the production of signal molecules called Nod factors. These signal molecules, which are *N*-acetyl-glucosamine oligomers exhibiting different chemical decorations [14], elicit different responses in the plant, which allows the start of bacterial infection and the beginning of nodule development. In addition to Nod factors, other bacterial signaling molecules, such as surface polysaccharides and effector proteins (called Nops, for nodulation outer proteins) delivered through a type 3 protein secretion system (T3SS), may play important roles in root infection and nodule cell invasion by rhizobia [13,15,16]. Although NodD is the master regulator of the different symbiotic genes whose expression is induced by flavonoids (the so called *nod* regulon), there are other regulatory proteins that may have a role in fine-tuning the expression of genes belonging to the *nod* regulon: TtsI (the positive activator of the symbiotic T3SS), SyrM (also belonging to the LysR family of transcriptional regulators), NolR (a helix-turn-helix negative regulator of nodulation genes), as well as additional NodD proteins (that may have both positive or negative influence on gene expression depending on the rhizobial species) [9,10,13,17,18,19,20].

In previous works, we have studied the regulation of the nodulation genes of *R. tropici* CIAT 899 and *S. fredii* HH103. Genome sequencing of CIAT 899 allowed the identification of five different *nodD* genes and three different *nodA* genes in the symbiotic plasmid [6]. Mutation of each *nodD* gene revealed that only *nodD1* is essential for nodulation with *Leucaena leucocephala*, *L. japonicus* and *Macroptilium atropurpureum*, but not with *P. vulgaris* and *L. burtii,* indicating the existence of a complex symbiotic regulatory circuit in this bacterium. Furthermore, the *nodD2* mutant induced fewer nodules in *P. vulgaris* than the wild-type strain, suggesting the putative implication of NodD2 in the interaction with this plant. Only the simultaneous inactivation of *nodD1* and *nodD2* abolished the ability to induce nodules in all host legumes assayed [21]. Moreover, CIAT 899 can produce considerable amounts of Nod factors in the absence of flavonoids when cultured under abiotic stress conditions such as acidity or high salt concentrations [22,23,24]. In fact, the presence of the inducer flavonoid or salt stress (300 mM) induced expression of a similar set of genes in CIAT 899 [25]. NodD1 is responsible for the biosynthesis of Nod factors in the presence of inducer flavonoids, whereas NodD2 and a regulator belonging to the AraC family activate Nod factor production under osmotic stress [21,26]. 

Regarding *S. fredii* HH103, NodD1 acts as the main and essential positive activator of the *nod* regulon [27], including other genes encoding secondary regulators such as *ttsI*, *nodD2*, and *syrM* [28,29]. As mentioned above, TtsI is the positive regulator of the symbiotic T3SS that is present in this strain [19]. Although the SyrM and NodD2 proteins activate the expression of a few genes belonging to the *nod* regulon, their predominant role is a repressor effect on the expression of genes involved in Nod factor production [17,18,30]. In addition, another transcriptional regulator, NolR [17,20], also has a role in decreasing the expression of HH103 nodulation genes. Interestingly, the inactivation of either *nodD2*, *syrM*, *ttsI* or *nolR* has a negative effect on the symbiotic performance of HH103 with soybean plants but extends the nodulation range of this strain to *Lotus japonicus* [17,18,31]. 

In this work, we show that the *nodD1* gene of HH103 restores Nod factor production and nodulation of a CIAT 899 *nodD1 nodD2* mutant in all the host legumes tested. We also show that the *nolR*, *nodD2* and *syrM* derivatives of *S. fredii* HH103 are not only able to induce the formation of nodule primordia in *Phaseolus vulgaris* roots, but also to establish an effective symbiotic interaction with this legume.

## 2. Materials and Methods

### 2.1. Microbiological Techniques

All strains and plasmids used in this work are listed in Appendix A. *R. tropici* CIAT 899 and *S. fredii* HH103 strains and derivatives were grown at 28 °C on tryptone yeast (TY) medium B^−^ minimal medium or yeast extract mannitol (YM) medium, supplemented, when necessary, with the inducer flavonoids apigenin or genistein 3.7 μM [32,33,34]. *Escherichia coli* strains were cultured on LB medium at 37 °C [35]. When required, the media were supplemented with the appropriate antibiotics as previously described [17]. Plasmids were transferred from *E. coli* to rhizobia by conjugation, as described by Simon [36].

### 2.2. Identification of Nod Factors and Biological Activity Assays

Nod factor purification and LC-MS/MS analyses were performed as previously described [9,10] by growing the bacteria in B^−^ minimal medium, supplemented when required with apigenin or genistein at a concentration of 3.7 μM. These experiments were carried out in duplicate and performed two independent times. Appendix A contains the list of Nod factors as well as their HPLC–HRMS signal areas. The purified Nod factors were used for biological activity assays (development of nodule primordia) as previously described [21]. These assays were performed three independent times, and at least 6 *P. vulgaris* plantlet roots were analyzed per treatment and assay.

### 2.3. Nodulation Assays 

For the evaluation of the symbiotic phenotypes, surface-sterilized seeds of *P. vulgaris* (cv. Blue Bush Lake)*, Lotus burttii* Borsos and *L. japonicus* Gifu were pre-germinated and placed on sterilized Leonard jars containing Farhaeus N-free solution [37], as previously described [17]. Germinated seeds were then inoculated with 1 mL of bacterial culture (about 10^8^ cells/mL). Growth conditions were 16 h at 26 °C in the light and 8 h and 18 °C in the dark, with 70% humidity. Nodulation parameters were evaluated after 30 days for beans and 50 days for *Lotus*. Shoots were dried at 70 °C for 48 h and weighed. Bacterial isolation from surface-sterilized nodules for analyzing the identity of bacteria occupying the nodules and ARA (acetylene reduction assay) for assessing nitrogenase activity of nodules were performed as previously described [38]. Nodulation experiments were performed three times. One representative experiment is shown in figures. 

## 3. Results

### 3.1. The nodD1 Gene of S. fredii HH103, but Not nodD2, Restores Nodulation Ability in a R. tropici CIAT 899 nodD1 nodD2 Mutant

As previously mentioned, inactivation of both *nodD1* and *nodD2* abolishes Nod factor production and, so, the ability to elicit formation of nodule primordia on *P. vulgaris* roots as well as the nodulation ability of *Rhizobium tropici* CIAT 899 with all the host legumes tested so far [21]. *Sinorhizobium fredii* harbors two different copies of *nodD*, called *nodD1* and *nodD2*, although only the former one appears to be directly involved in the activation of genes involved in Nod factor production [17,27]. In order to investigate whether the *nodD1* or *nodD2* genes from *S. fredii* HH103 may restore nodulation ability in a *R. tropici* CIAT899 Δ*nodD1* Δ*nodD2* mutant (hereafter CIAT 899 Δ*nodD1D2*), we have introduced in this strain plasmids pMUS296 and pMUS746 carrying the *nodD1* and *nodD2* genes from HH103, respectively [19,27].

First, we investigated the symbiotic properties of CIAT 899 Δ*nodD1D2* carrying either the *nodD1* or the *nodD2* genes from *S. fredii* HH103 in three different host plants: *P. vulgaris*, *L. burttii* and *L. japonicus* (Figure 1). Strains CIAT 899, CIAT 899 Δ*nodD1D2*, and HH103 were also included in these experiments as controls. Strain CIAT 899 Δ*nodD1D2* carrying the empty vector pMP92 behaved as CIAT 899 Δ*nodD1D2* in the three legumes tested. Three different parameters were analyzed for *P. vulgaris*: number of nodules, fresh mass of nodules (FMN) and stem dry biomass (SDB). The first two parameters inform about the efficiency of nodulation, whereas the latter parameter is a good indication of nitrogen fixation efficiency. In the case of both species of *Lotus*, the number of nodules and SDB were scored at the end of the experiment. As expected, CIAT 899 induced the formation of nitrogen-fixing nodules (Fix^+^) in the three host plants tested, whereas CIAT 899 Δ*nodD1D2* failed to nodulate these legumes (not even pseudonodules). *S. fredii* HH103 was effective on *L. burttii*, but only elicited the formation of pseudonodules (empty nodules unable to fix nitrogen) in both *L. japonicus* and *P. vulgaris*. The presence of pMUS296, but not that of pMUS746, allowed CIAT 899 Δ*nodD1D2* to induce the formation of nitrogen fixing nodules in the three host legumes tested: *P. vulgaris*, *L. burttii* and *L. japonicus*. In *Lotus burttii*, the number of Fix^+^ nodules induced by CIAT 899 and CIAT899 Δ*nodD1D2* (pMUS296) and the FMN of nodules were similar, but *P. vulgaris* and *L. japonicus* plants inoculated with the complemented mutant formed approximately half of the Fix^+^ nodules (with a significantly lower FMN) when compared to those inoculated with CIAT 899. Regarding SDB, plants inoculated with CIAT 899 Δ*nodD1D2* (pMUS296) exhibited values that were significantly higher than those of non-inoculated plants but lower than those inoculated with CIAT 899. The results obtained prompted us to investigate the competitive ability of CIAT 899 Δ*nodD1D2* (pMUS296) to nodulate beans. When inoculated at a 1:1 ratio with CIAT 899, the complemented mutant strain only occupied the 18.5 ± 4.6 of the formed nodules.

Since CIAT 899 Δ*nodD1D2* (pMUS296) is able to effectively nodulate bean and *Lotus*, we analyzed the Nod factor production by this strain upon induction with apigenin following the methodology detailed in the Material and Methods Section. As shown in Figure 2 and Appendix A, the presence of the NodD1 protein of HH103 in a CIAT 899 Δ*nodD1D2* genetic background, unable to produce Nod factors [21], was enough to restore the synthesis and export of 63 out of 99 LCOs naturally occurring in *R. tropici* CIAT 899 in the presence of apigenin, indicating that this NodD1 protein is transcriptionally activating the nodulation genes of CIAT 899 under inducing conditions. We have not found a clear pattern of differences between the sets of Nod factors produced by these strains regarding specific modifications. Both strains produced Nod factors with 3, 4 and 5 *N*-acetyl-glucosamine residues and harboring sulfate, *N*-methylations, and/or carbamoyl residues, as well as different kinds of fatty acids. Similar results have been previously obtained when the sets of Nod factors produced by CIAT 899 and individual mutants in either *nodD1* or *nodD2* were compared [9].

### 3.2. Inactivation of the Symbiotic Regulators nodD2, syrM or nolR Extends the Nodulation Range of S. fredii HH103 to Phaseolus vulgaris

The fact that *S. fredii* HH103 can induce the formation of pseudonodules on *P. vulgaris* roots suggests that the Nod factors produced by this strain might be perceived by the plant and trigger the nodule developmental program, but are not able to allow root infection and/or nodule invasion. In order to study this possibility, the Nod factors produced by CIAT 899 (as the positive control) and HH103 were purified as described in the Material and Methods and following induction with the appropriate flavonoid (apigenin for CIAT 899 and genistein for HH103). The purified Nod factors were applied on roots of *P. vulgaris* plantlets, and the appearance of nodule primordia was scored 10 days after treatment. Appendix A shows the aspect of nodule primordia on *P. vulgaris* roots, that are rounded and distinguishable from emerging lateral roots, which are conical. Non-inoculated roots of *P. vulgaris* plantlets were also analyzed as a negative control of the appearance of nodule primordia. Nod factors from both CIAT 899 and HH103 were able to elicit nodule organogenesis, since in both cases nodule primordia could be found, although Nod factors from CIAT 899 were more efficient than those of HH103 (20.3 ± 5.5 primordia per root vs. 9.7 ± 1.5, respectively).

In a previous work of our group, we showed that strain HH103 induced the formation of white, non-colonized nodules, in the model plant *Lotus japonicus* [12]. However, HH103 mutants affected in different “secondary” regulators of nodulation genes (*nodD2*, *nolR*, *syrM*, *ttsI*) gained the ability to infect *L. japonicus* roots and induce the formation of Fix^+^ nodules. For this reason, we decided to study whether a similar result could be obtained with *P. vulgaris*. Thus, we have investigated the symbiotic performance of HH103 mutants in these regulatory genes with beans plant (Figure 3, Figure 4 and Appendix A).

The *ttsI* derivative of HH103, as the wild-type strain, only induced pseudonodules and non-fully developed nodules that were white and unable to fix nitrogen as assessed by ARA (acetylene reduction assay), resulting in plants showing nitrogen starvation symptoms and a stem dry biomass (SDB) similar to that of non-inoculated plants. No bacteria could be isolated from the structures induced by either HH103 or its *ttsI* mutant. The HH103 *syrM* mutant was able to induce the formation of well-formed nodules. However, most of them were white and ARA analyses reflected a very week nitrogenase activity per nodule (90.6 ± 19.9 nmol ethylene per hour). In addition, the SDB was also not significantly different to that of non-inoculated plants. Finally, the *nolR* or *nodD2* mutants of HH103 were both able to induce the formation of nitrogen-fixing nodules in *P. vulgaris*, as CIAT 899 did. The comparison of mutants from one strain (HH103) with another wild-type strain (CIAT 899) may not be conclusive. However, it is remarkable that the number of nodules formed by plants inoculated with the *nolR* and *nodD2* mutants of HH103 was approximately half of that developed by plants inoculated with CIAT 899. This fact was reflected in a lower nodule fresh mass in the case of HH103 *nolR*, but not for HH103 *nodD2* (for this strain, the value of this parameter was indistinguishable from that of CIAT 899). ARA analyses showed a weaker nitrogenase activity in plants inoculated with either HH103 *nodD2* (408.5 ± 102.5 nmol ethylene per hour per nodule) or HH103 *nolR* (239.1 ± 32.7) when compared to those plants inoculated with CIAT 899 (806.4 ± 53.0). Nevertheless, for those plants inoculated with either the *nodD2* or the *nolR* mutant of HH103, the SDB was significantly higher than that of non-inoculated plants, but significantly lower than that of plants inoculated with CIAT 899.

## 4. Discussion

One of the main characteristics of the symbiotic interaction between rhizobia and legumes is specificity [3,5,13]: each rhizobial strain is able to induce the nodulation program, to infect the roots and to colonize the nodules of a specific set of legumes. The accomplishment of all these events depends on the ability of the bacterium to recognize flavonoids and other molecules exuded by the plant and to produce appropriate molecular signals that trigger nodule organogenesis and allow the infection and colonization processes.

Rhizobial NodD proteins interact with plant flavonoids and, when the interaction is appropriated, trigger the expression of bacterial genes responsible for Nod factor production and secretion [39,40]. The NodD proteins of different rhizobia differ in their ability to recognize flavonoids, which justifies partially the fact that some rhizobia (such as *S. meliloti*) have narrow host-ranges whereas other rhizobia (such as *S. fredii*) can interact effectively with many legumes [27,41]. *S. fredii* strains harbor two different copies of *nodD*, *nodD1* and *nodD2* [13]. NodD1 is the positive regulator of the *nod* regulon, being able to interact with a great diversity of flavonoids [27] and, in response, to trigger the expression of several dozens of genes, including those involved in Nod factor production but also some regulatory genes such as *ttsI*, *nodD2* or *syrM* [28]. Different reports suggest that *S. fredii* NodD2, instead, acts as a repressor that reduces the level of expression of genes involved in Nod factor production [17,30,42]. The results obtained in this work agree with those previous observations, since the *nodD1* gene of HH103, but not *nodD2*, was able to drive the production of Nod factors in a *R. tropici* CIAT 899 Δ*nodD1D2* mutant upon induction with apigenin, although there were some differences in the variety and relative abundances of the different Nod factors produced (Figure 2; Appendix A). The presence of the *nodD1* gene of HH103, but not that of *nodD2*, also restored the ability of this strain to nodulate the three different host plants of *R. tropici* tested in this work (bean, *L. burttii* and *L. japonicus*). However, in all these legumes the symbiotic performance of CIAT 899 Δ*nodD1D2* (pMUS296) was significantly lower (as estimated by scoring nodule number and/or SDB) to that of CIAT 899 (Figure 1), which suggests that the set of Nod factors produced in the presence of the *nodD1* gene of HH103 was more inefficient to promote nodule organogenesis (as also suggested by the analysis of nodule primordia formation) and/or root infection and nodule invasion than that of CIAT 899. 

So far, the number of CIAT 899 genes known to be involved in Nod factor production is small: *nodA1BCSUIJH*, *nodA2hsnTnodFE*, and *nodM* [21]. The products of these genes are responsible for synthesis of the Nod factors backbone (NodABC), *N*-methylation (NodS), carbamoylation (NodU), addition of sulfate (NodH), unsaturated fatty acid incorporation (HsnTNodFE), synthesis of glucosamine (NodM), and Nod factor export (NodIJ). All these *nod* box dependent genes appear to be induced in CIAT 899 Δ*nodD1*/*nodD2* carrying the HH103 *nodD1* gene (in the presence of flavonoids) since all these modifications can be found in the Nod factors produced by this strain.

In addition to the positive role of NodD and flavonoids, the expression of rhizobial symbiotic genes is finely modulated in a species- (or even strain-) specific manner and involves the participation of different “secondary regulators” such as additional copies of *nodD*, *syrM* or *nolR* [5,17,18,43]. In previous works, we have shown that *S. fredii* HH103 mutants affected in *nodD2*, *syrM* or *nolR* produced similar sets of Nod factors as the parental strain but in higher amounts than the latter strain [17,18]. Interestingly, these three mutants gained the ability to fix nitrogen on *Lotus japonicus*, whereas the wild-type strain only induced pseudonodules devoid of bacteria. In this work we demonstrated a similar result regarding *Phaseolus vulgaris* (Figure 3, Figure 4 and Appendix A): these three mutants, in contrast to the pseudonodules induced by HH103, led to the formation of nitrogen-fixing nodules, although only the *nodD2* and *nolR* mutants exhibited nitrogen fixation levels that resulted in an improvement in plant development (scored as SDB) when compared to non-inoculated plants. The sets of HH103 genes whose expression is affected by inactivation of *nodD2*, *nolR* or *syrM* are large (several hundred each) and different [17,18], so it is really difficult to elucidate the reasons of the differences in symbiotic performance of these three mutants with *Phaseolus*. However, it is remarkable that the *ttsI* derivative of HH103 (unable to secret Nops) exhibited the same symbiotic phenotype with *Phaseolus* as the wild-type strain (formation of pseudonodules), whereas this mutant also gained the ability to effectively nodulate *Lotus japonicus* [31]. Although this result suggests that HH103 Nops prevent nodulation on *L. japonicus* but not on *P. vulgaris*, further studies should be done for analyzing the effect of the *ttsI* mutation in HH103 derivatives carrying additional mutations in either *nodD2*, *nolR* or *syrM*.

As mentioned above, regulation of the production of symbiotic molecules is complex and vary among different rhizobia. In this work we provide additional evidence of this fact. In *S. fredii* HH103, NodD1 is able to strongly induce *nod* gene expression in response to most of the flavonoids and radical exudates tested so far [17,27]. However, this HH103 protein was not able to fully complement a *R. tropici* CIAT 899 Δ*nodD1D2* mutant. In fact, the symbiotic behavior of this mutant carrying the HH103 *nodD1* gene is similar to that previously observed for CIAT 899 Δ*nodD1D2* carrying the wild type *nodD1* gene [21], suggesting that the presence of the HH103 *nodD1* gene might complement the lack of CIAT 899 NodD1 but not that of CIAT 899 NodD2. On the other hand, the enhanced production of Nod factors caused by inactivation of *syrM*, *nolR,* or *nodD2* in HH103 is most probably the reason why these mutants gained the ability to induce the formation of Fix^+^ nodules on two legumes*, L. japonicus* and *P. vulgaris*, in which the parental strain is ineffective. Interestingly, all these mutants were significantly less efficient than HH103 in the symbiotic interaction with soybean, the main partner of *S. fredii*. In our opinion, this fact reflects that the complex regulation of rhizobial symbiotic genes has evolved in order to optimize the interaction with the habitual host legumes, although it may avoid interaction with other legumes.

## Figures and Tables

**Figure 1 microorganisms-10-00139-f001:**
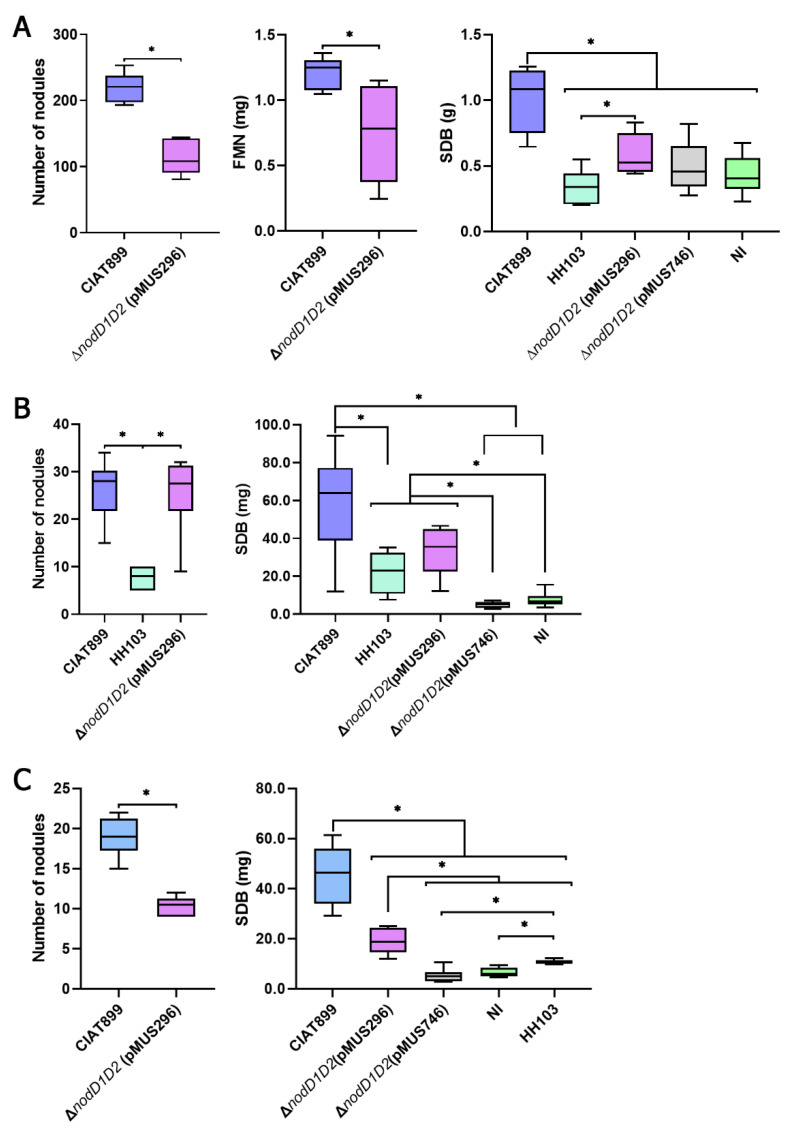
Symbiotic performance of *S. fredii* HH103, *R. tropici* CIAT 899, and CIAT 899 Δ*nodD1D2* carrying the *nodD1* (pMUS296) or the *nodD2* (pMUS746) gene of HH103 in *Phaseolus vulgaris* (panel (**A**)), *Lotus burttii* (panel (**B**)), and *L. japonicus* (panel (**C**)). FMN, fresh mass of nodules. SDB, stem dry biomass. All treatments were compared to each other following the Mann–Whitney non-parametric test. Significant differences (α = 0.5%) are denoted by asterisks.

**Figure 2 microorganisms-10-00139-f002:**
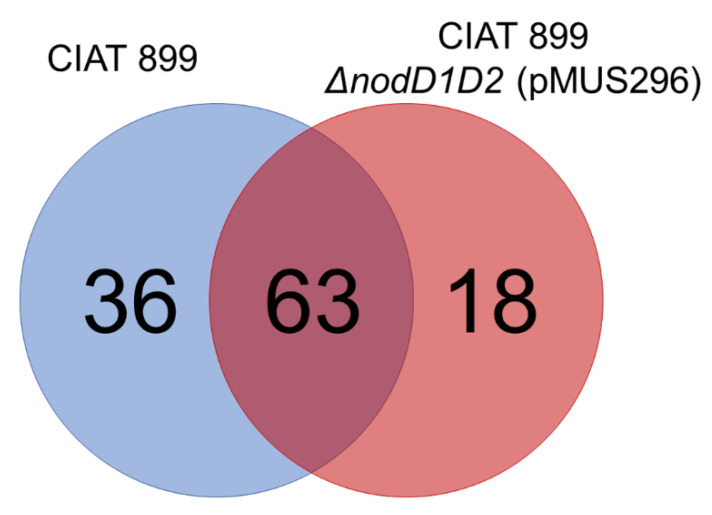
Comparative analysis of the set of Nod factors produced by *R. tropici* CIAT 899 (blue circle) and CIAT 899 Δ*nodD1D2* (pMUS296) (red circle). The differences among these strains are visualized by a Venn diagram.

**Figure 3 microorganisms-10-00139-f003:**
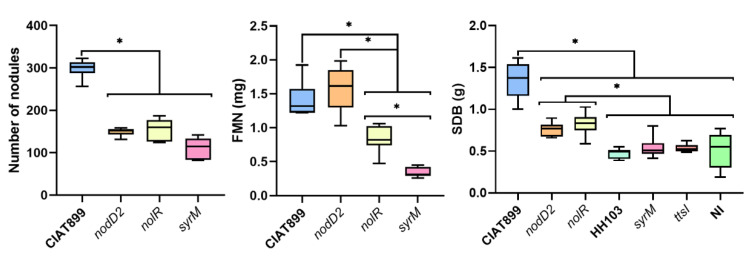
Symbiotic performance of *R. tropici* CIAT 899, *S. fredii* HH103 and its *nodD2*, *nolR*, *syrM*, and *ttsI* derivatives with *Phaseolus vulgaris*. FMN, fresh mass of nodules. SDB, stem dry biomass. All treatments were compared to each other following the Mann–Whitney non-parametric test. Significant differences (α = 0.5%) are denoted by asterisks.

**Figure 4 microorganisms-10-00139-f004:**
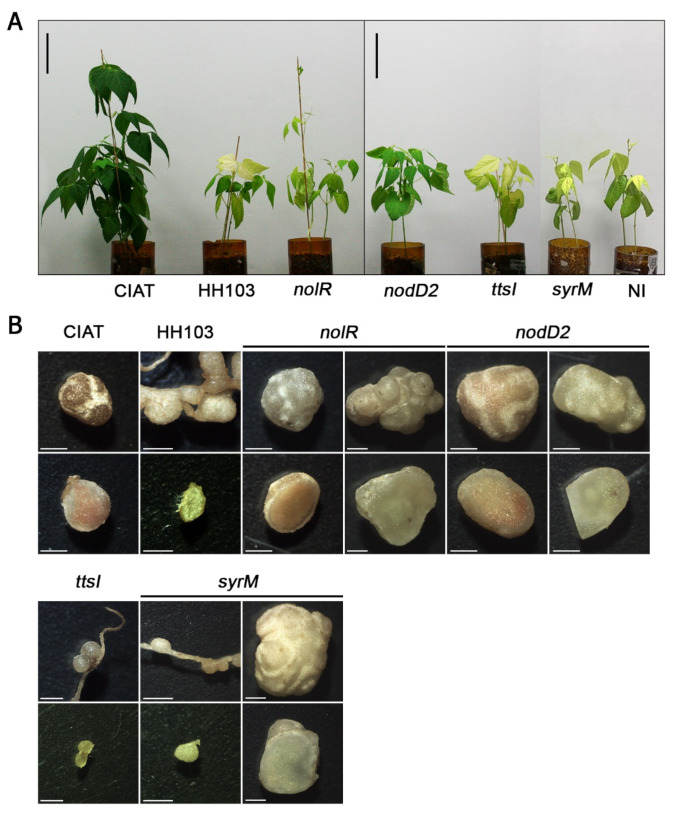
Symbiotic performance of *R. tropici* CIAT 899, *S. fredii* HH103 and its *nodD2*, *nolR*, *syrM*, and *ttsI* derivatives with *Phaseolus vulgaris*. Appearance of aerial parts (panel (**A**)), and nodule and pseudonodule morphology (panel (**B**)). Bars represent 10 cm for panel (**A**) and 1 mm for panel (**B**).

## Data Availability

The list of Nod Factors produced by *R. tropici* CIAT 899, CIAT 899 Δ*nodD1D2*, and CIAT 899 Δ*nodD1D2* (pMUS296) are available as Dataset S1.

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
