# Peer review of "The nodD1 Gene of Sinorhizobium fredii HH103 Restores Nodulation Capacity on Bean in a Rhizobium tropici CIAT 899 nodD1/nodD2 Mutant, but the Secondary Symbiotic Regulators nolR, nodD2 or syrM Prevent HH103 to Nodulate with This Legume"

_microorganisms, 2022, doi:10.3390/microorganisms10010139_

Round 1

Reviewer 1 Report

Review of ms Fuentes-Romero et al. submitted to Microorganisms

This paper provides information on the complex effects of rhizobial regulatory genes influencing Nod factor synthesis. The authors show that introduction of the nodD1 gene from Sinorhizobium fredii HH103 into a previously constructed nodD1/ nodD2 double mutant of CIAT899 can partially restore the nodulation capacity in bean plants. Nod factor analysis by mass spectrometry indicated certain differences in the complementation strain as compared to CIAT899 wildtype. Furthermore, nodD2, syrM and nolR knockout mutants of HH103 with increased Nod factor production positively affected nodulation of beans as previously shown for Lotus japonicus.

3 nodD1/nodD2

14  Background information is missing. Explain first what NodD etc means.

16 in bean  plants.

18 protein secretion

24-25 Nod factor production (also elsewhere in text)

36 with carbon and nutrients. The

37 commonly referred as

41 beneficial for sustainable

43 genera? Sometimes there is specificity for a distinct cultivar/ecotype of the same species. Consider modification such as “specific host legumes”

52 dialogue, what explains, at

58 Nod factors

58 N-acetyl (write N in italics)

59 elicit

63 protein secretion

66-67  a role in fine-tuning the expression of genes belonging

70 additional NodD proteins

75 essential for nodulation with

78 Furthermore, the

78 induced fewer nodules in

80-81 Delete “mutant”

82 can produce considerable amounts of

82 when cultured under

83 high salt concentrations

84 induced expression of a similar set of

87 factor

89 including genes encoding secondary regulators such as TtsI, NodD2 and SyrM

93 factor

98 factor

113 Nod factors

116 at a concentration of 3.7 µM.

116 Nod factors

121 Which bean genotype was used? Cultivar? Seed source?

133 factor

135 factor

140 What means “Fitas”?

145 “, respectively” should be written in the end of a sentence

143-145 Which promoters? Native promoters? How many bp upstream of the ATG start codon?

151 I my opinion, “plant tops” is rarely used and should be rather replaced by “above-ground biomass” or “stem biomass”

152 fixation efficiency.

153 “Lotus” in italics

169 occupied 18.5 ± X.X% of the formed nodules.

Fig. 1 Number of nodules

Fig. 1 What is the phenotype of the double mutant carrying a plasmid with the nodD1 gene of CIAT 899? Is the difference between CIAT and the shown complementation strain due to differences in the two NodD1 proteins?  How different are these two NodD1 proteins?  Are there plasmid effects (no empty vector control shown). Are the used plasmids stable? (do all bacteria re-isolated from nodules contain the plasmid?).

Fig. 2 Which Nod factors are unique for CIAT 899 and the complementation strain? Consider moving data (Nod factor structures) from the Supplementary Information to here.

263 10 days after treatment.

275 decided

277 bean plants

281 See comments line 151. Explain abbreviations when mentioned first, then use abbreviations. In my opinion, avoid abbreviations rarely used in the literature such as “PTDM” and “FMN” and instead use commonly used expression such as “Stem biomass (DW)”.

290 In this sentence important. Comparisons of mutants from one strain with another wild-type strain are not rather conclusive.

293 See comments line 290

293 Replace “In any case” by another word.

313 of the symbiotic interaction between rhizobia and legumes

313-318 This general part should be rather integrated into the Introduction

339 Which specific set of Nod factors are different (see comments to Fig. 2). Discuss which nod genes would be involved in synthesis of these specific Nod factors.

392 mutant derivatives

396 See comment to 339. Would analysis of these genes and the structure of the different Nod factors shown in Fig. 2 provide any information on production of Nod factors with specific modifications.

410 Here you come up that Nod factor amounts are “most probably” important for symbiosis and not “specific Nod factor structures”.  Why “most probably”?  In fact, Nod factor levels of CIAT 899 and the complementation strain were not quantified in this study. The results presented in this study remain therefore somehow “patchy” and are difficult to interpret. This should be better emphasized in the Discussion. The unsolved question of “Nod factor structures” versus “Nod factor levels” could be better discussed.

409 Explain “nod boxes” in the Introduction. Besides genes involved in Nod factor production and T3SS genes, are there any other genes in HH103 with “nod boxes”?

Author Response

Microorganisms-1532292, Cover letter of the revised version of the manuscript

Dear Ms. Lily Liang,

We have sent our revised version of manuscript ID microorganisms-1532292 (“The nodD1 gene of Sinorhizobium fredii HH103 restores nodulation capacity on bean in a Rhizobium tropici CIAT 899 nodD1 nodD2 mutant, but the secondary symbiotic regulators nolR, nodD2 or syrM prevent HH103 to nodulate with this legume”).

We would like to sincerely thank again the two reviewers for their really useful comments and suggestions and for carrying out their work so quickly. In our opinion, they have contributed to improve the quality of the manuscript.

These are our answers to the different questions posed by the reviewers.

Reviewer 1

Review of ms Fuentes-Romero et al. submitted to Microorganisms

This paper provides information on the complex effects of rhizobial regulatory genes influencing Nod factor synthesis. The authors show that introduction of the nodD1 gene from Sinorhizobium fredii HH103 into a previously constructed nodD1/ nodD2 double mutant of CIAT899 can partially restore the nodulation capacity in bean plants. Nod factor analysis by mass spectrometry indicated certain differences in the complementation strain as compared to CIAT899 wildtype. Furthermore, nodD2, syrM and nolR knockout mutants of HH103 with increased Nod factor production positively affected nodulation of beans as previously shown for Lotus japonicus.

3 nodD1/nodD2

Done

14 Background information is missing. Explain first what NodD etc means.

Done

16 in bean plants.

Done

18 protein secretion.

Done

24-25 Nod factor production (also elsewhere in text).

Done

36 with carbon and nutrients. The…

Done

37 commonly referred as…

Done

41 beneficial for sustainable…

Done

43 genera? Sometimes there is specificity for a distinct cultivar/ecotype of the same species. Consider modification such as “specific host legumes”

Reviewer is right. Thank you for this suggestion.

52 dialogue, what explains, at…

Done

58 Nod factors…

Done

58 N-acetyl (write N in italics).

Done

59 elicit.

Done

63 protein secretion.

Done

66-67  a role in fine-tuning the expression of genes belonging…

Done

70 additional NodD proteins.

Done

75 essential for nodulation with…

Done

78 Furthermore, the…

Done

78 induced fewer nodules in…

Done

80-81 Delete “mutant”…

Done

82 can produce considerable amounts of…

Done

82 when cultured under…

Done

83 high salt concentrations…

Done

84 induced expression of a similar set of…

Done

87 factor…

Done

89 including genes encoding secondary regulators such as TtsI, NodD2 and SyrM…

Done

93 factor.

Done

98 factor.

Done

113 Nod factors.

Done

116 at a concentration of 3.7 µM.

Done

116 Nod factors.

Done

121 Which bean genotype was used? Cultivar? Seed source?

The bean cultivar is Blue Bush Lake. These seeds are available at most seed shops. We have also added the L. burttii cultivar employed in this work.

133 factor.

Done

135 factor.

Done

140 What means “Fitas”?

We apologize for this mistake: when we changed the references to the “number format” we forgot to replace two of them. FITA is the acronym for Flavonoid Independent Transcription Activation and refers to spontaneous S. fredii nodD1 mutants able to induce nod gene expression in the absence of flavonoids. The study of this kind of mutants is included in our work describing the nodD1 gene of S. fredii HH103 (reference 27, that remained in the text as Vinardell et al. 2004fitas).

145 “, respectively” should be written in the end of a sentence.

Done

143-145 Which promoters? Native promoters? How many bp upstream of the ATG start codon?

Construction of plasmids pMUS296 and pMUS741 is described in references 27 and 19, respectively. In both cases the vector is the broad host-range IncP plasmid pMP92 (Spaink HP, Okker RJH, Wijffelman CA, Pees E, Lugtenberg BJJ (1987) Promoters in the nodulation region of the Rhizobium leguminosarum Sym plasmid pRL1JI. Plant Mol Biol 9:27–39). In both cases, the subcloned fragment carries the complete gene and approximately 300 bp of the upstream region.

151 I my opinion, “plant tops” is rarely used and should be rather replaced by “above-ground biomass” or “stem biomass”.

According to reviewer suggestion, we have replaced plant tops dry mass for stem dry biomass (SDB) in text and figures.

152 fixation efficiency.

Done

153 “Lotus” in italics.

Done

169 occupied 18.5 ± X.X% of the formed nodules.

Corrected.

Fig. 1 Number of nodules.

Done

Fig. 1 What is the phenotype of the double mutant carrying a plasmid with the nodD1 gene of CIAT 899? Is the difference between CIAT and the shown complementation strain due to differences in the two NodD1 proteins?  How different are these two NodD1 proteins?  Are there plasmid effects (no empty vector control shown)? Are the used plasmids stable? (do all bacteria re-isolated from nodules contain the plasmid?).

Thank you for your very interesting comments.

The symbiotic phenotype in bean and in L. burttii of the double mutant carrying a plasmid with the nodD1 gene of CIAT 899 has been described by del Cerro et al. (2017) (reference 21) and it is similar to the symbiotic phenotype exhibited by the double mutant carrying the nodD1 gene of S. fredii HH103: a reduction in the number of nodules formed (although only significant in the case of bean) and reduction in stem dry biomass (although not significant at α=5%). In fact, the phenotypes of the double mutant carrying a nodD1 gene (either of CIAT 899 or of HH103) are similar to that of a CIAT 899 nodD2 mutant. Because of this, we do not think that the difference between CIAT and the shown complementation strain may be due to the lack of NodD2 rather than to differences in the two NodD1 proteins. The NodD1 proteins of CIAT 899 (accession number AGB73531) and HH103 (accession number CEO91589) are 73% identical. But the critical point is that HH103 NodD1 is able to induce nod gene expression driven by a heterologous nod box (the nodA promoter from Rhizobium leguminosarum present in plasmids pMP240 and pMP154; please, see reference 27 for further details) in the presence of most of the flavonoids tested (those carrying a hydroxyl group at C7). Thus, the HH103 NodD1 protein is able to induce nod gene expression in the presence of a broad spectrum of flavonoids and root exudates, which, partially justify the broad host-range of S. fredii strains.

Of course, the control with empty vector (pMP92) has been carried out: the double mutant carrying pMP92 behaved as the double mutant, being fully impaired in symbiosis with the different host legumes assayed. We have added this information to the ms (see lines 156-158).

Plasmid pMP92 (the TcR vector used for construction of plasmids pMUS296 and pMUS746) is not fully stable in the absence of selective pressure. In those experiments performed for obtaining Nod factors, cultures were supplemented with tetracycline, so all cells in the culture should maintain the plasmid. Regarding nodulation assays, bacterial cultures used as inocula were grown in the presence of tetracycline, washed and added to plantlet roots. More than 75% of the analysed nodule isolates were TcR.

Fig. 2 Which Nod factors are unique for CIAT 899 and the complementation strain? Consider moving data (Nod factor structures) from the Supplementary Information to here.

This is a very interesting question. We have not found a clear pattern of differences between the sets of Nod factors produced by these strains regarding specific modifications. Both strains produced Nod factors with 3, 4 and 5 N-acetyl-glucosamine residues and harboring sulfate, N-methylations, and/ or carbamoyl residues, as well as different kind of fatty acids. Similar results have been previously obtained when the sets of Nod factors produced by CIAT 899 and individual mutants in either NodD1 or nodD2 were compared (reference 9). In our opinion, our results suggest that the presence of pMUS296 complement the lack of CIAT 899 NodD1 but not that of CIAT 899 NodD2. In fact, the symbiotic phenotypes of CIAT899 nodD1/nodD2 harboring pMUS296 are very similar to that of CIAT899 nodD1/nodD2 harboring the nodD1 gene of CIAT899. In addition, to relate specific Nod factors with the different symbiotic phenotypes exhibited by CIAT 899 and CIAT899 nodD1/nodD2 harboring pMUS296 would be a very complicated task that would require further research.

We commented on this issues in the Results and in Discussion sections of the revised ms (please, see lines 186-192 and 344-353).

263 10 days after treatment.

Done

275 decided.

Done

277 bean plants.

Done

281 See comments line 151. Explain abbreviations when mentioned first, then use abbreviations. In my opinion, avoid abbreviations rarely used in the literature such as “PTDM” and “FMN” and instead use commonly used expression such as “Stem biomass (DW)”.

Done.

290 In this sentence important. Comparisons of mutants from one strain with another wild-type strain are not rather conclusive. 293 See comments line 290

Reviewer is right. We only mean to clearly state that the analyzed HH103 mutants have little symbiotic efficiency when compared to a typical symbiont of Phaseolus vulgaris, CIAT 899. And this is the only comparison that we can make since HH103 is not able to induce the formation of nitrogen-fixing nodules in this plant. We have modified this part of the manuscript accordingly to reviewer’s comments.

293 Replace “In any case” by another word.

Done, we replace “in any case” by “anyway”.

313 of the symbiotic interaction between rhizobia and legumes.

Done.

313-318 This general part should be rather integrated into the Introduction.

Thank you for your comment. We would like to leave this little paragraph here as a small introduction for the role of the different symbiotic regulators in determining host specificity.

339 Which specific set of Nod factors are different (see comments to Fig. 2). Discuss which nod genes would be involved in synthesis of these specific Nod factors.

The number of CIAT 899 genes known to be involved in Nod factor production is small: nodA1BCSUIJH, nodA2hsnTnodFE and nodM (and also a third copy of nodA that is constitutively expressed). The products of these genes are responsible for synthesis of the backbone (NodABC), N-methylation (NodS), carbamoylation (NodU), addition of sulfate (NodH), unsaturated fatty acid incorporation (HsnTNodFE), synthesis of glucosamine (NodM), and Nod factor export (NodIJ). All these nod box dependent genes appear to be induced in CIAT 899 ΔnodD1/nodD2 carrying the HH103 nodD1 gene since all these modifications can be found in the Nod factors produced by this strain. Thus, the presence of HH103 NodD1 in CIAT 899 appears to induce (in the presence of flavonoids) all the nodulation genes of CIAT 899 identified so far.

We have commented on this issue in the Discussion (please, see lines 394-400).

392 mutant derivatives.

Done.

396 See comment to 339. Would analysis of these genes and the structure of the different Nod factors shown in Fig. 2 provide any information on production of Nod factors with specific modifications.

Thank you for your comment. The sets of Nod factors produced by the HH103 nolR, nodD2 and syrM mutant derivatives has been previously studied (please, see references 17 and 18). In all these three mutants the sets of Nod factors produced were basically the same as that of the wild-type strain, being the most predominant tri-, tetra- and pentameric oligochitin skeletons carrying a C18:1, C18:0 or C16:1 fatty acid and a (methylated-) fucosyl residue. The big difference is the amount of Nod factors produced. All these three mutants produced increased amounts of many of these Nod factors (between 2 and 30 times more). Because of this, our hypothesis is that, as we discussed in references 17 and 18 for Lotus japonicus, it might be possible that some particular HH103 Nod-factors are inefficiently perceived by the Nod-factor receptors of Phaseolus vulgaris roots but that the presence of higher levels of the same signal molecules in the nodD2 nolR and syrM mutants could trigger the start of the nodulation and infection processes in this plant.

410 Here you come up that Nod factor amounts are “most probably” important for symbiosis and not “specific Nod factor structures”.  Why “most probably”?  In fact, Nod factor levels of CIAT 899 and the complementation strain were not quantified in this study. The results presented in this study remain therefore somehow “patchy” and are difficult to interpret. This should be better emphasized in the Discussion. The unsolved question of “Nod factor structures” versus “Nod factor levels” could be better discussed.

Thank you again for your comment. In our opinion, we have to differentiate two different issues:

  • The nodD1 gene of HH103 is able to drive nod gene expression in a CIAT 899 nodD1/nodD2 mutant (unable to produce Nod factors and with a Nod- phenotype in all the host legumes tested) leading to Nod factor production and the ability to effectively nodulate different host legumes of CIAT 899. In this case we are comparing Nod factors production by CIAT 899 and its nodD1/nodD2 mutant carrying the HH103 nodD1 gene. The fact that the CIAT 899 nodD1/nodD2 mutant carrying pMUS296 produced a lower number of different Nod factors than CIAT 899 is not strange since a CIAT nodD2 mutant also did (see reference 9). Thus, our results suggest that we are only complementing the nodD1 mutation but not the nodD2 one. We are not saying that CIAT 899 nodD1/nodD2 pMUS296 showed enhanced production of Nod factors, but that this strain is able to produce many of the CIAT 899 Nod factors, which allows nodulation in the different host legumes tested in this work.
  • The HH103 nodD2, nolR and syrM mutants produce higher amounts of Nod factors than the parental strain, and basically produce the same Nod Factors as HH103 does. This has been previously demonstrated (references 17 and 18). Because of this, our hypothesis is that this enhanced Nod factor production might be the cause of gaining nodulation ability in legumes, such as Lotus japonicus or Phaseolus vulgaris, that are not efficiently nodulated by the parental strain.

We have modified the Discussion in order to make clearer these messages. Please, see lines 405-408 and 429-434.

409 Explain “nod boxes” in the Introduction.

Done (please, see lines 57-60).

Besides genes involved in Nod factor production and T3SS genes, are there any other genes in HH103 with “nod boxes”?

Yes, they are. You can see reference 28 for more details. In fact, only two nod boxes control the expression of genes related to Nod factor production. Other nod boxes control the expression of regulatory genes (such as syrM or ttsI), genes involved in indole-3-acetic acid biosynthesis, genes related to nitrogen fixation, genes coding for hypothetical proteins, etc.).

Reviewer 2

In this paper, Fuentes-Romero and collaborators have analyzed the role of 5 symbiotic regulators (NodD1, NodD2, NolR, SyrM and TtsI) of S. fredii HH103, the natural symbiont of soybean but unable to nodulate efficiently L. japonicus and P. vulgaris. In previous works, authors showed that NodD1 is the main activator of nod genes in S. fredii, while NodD2, NolR and SyrM mainly act as repressors of the expression of nod genes and TtsI is a positive regulator of the T3SS of S. fredii. Moreover they previsouly showed that inactivation of the secondary regulators nodD2syrMnolR and ttsI allowed S. fredii to nodulate and infect L. janonicus. In this study, Fuentes-Romero and collaborators analyzed the role of these regulators on the symbiosis with P. vulgaris, another non-host plant of S. fredii, to see whether they play a negative role in this interaction like on L. japonicus.

Authors conducted this study in two ways. First, they analyzed the role of NodD1 (activator of nod genes) and nodD2 (repressor of nod genes) on the capacity to provide the nodulation ability of a double nodD1nodD2 mutant of R. tropici CIAT 899 a nitrogen-fixing symbiont of P. vulgaris. Introduction of nodD1, but not nodD2, on a plasmid allowed the nodulation of this strain on P. vulgaris (as well as on L. japonicus and L. Burtii) although with less efficiently than the wild-type strain, meaning that the S. fredii HH103 NodD1 is able to activate the nodulation genes of R. tropici CIAT899. Whether this lesser efficiency is due to the production of a different set of Nod Factors needs to be confirmed (see my comment below).

Besides, they analyze the role of the other regulators TtsI, NodD2, NolR and SyrM in S. fredii by constructing individual mutants of these genes in S. fredii HH103 and testing their nodulation and nitrogen fixation capacities on P. vulgaris. Authors found that a single tssI mutant does not allow nodulation on P. vulgaris while single mutants in nodD2nolR or syrM partially restored nodulation on P. vulgarisnodD2 and nolR mutants also partially restore the nitrogen fixation activity, indicating that these regulators negatively impact the interaction with P. vulgaris.

To strengthen the conclusions of the paper, I suggest some revisions.

First, I am wondering whether the effect of the ttsI inactivation is possible to observe in a strain that is not producing sufficient amounts of Nod Factors (inhibited by NodD2, NolR and SyrM). To reliably conclude that TtsI is not involved in host range restriction on P. vulgaris, the effect of the ttsI mutation should be tested in a nodD2 or nolR mutant or in a double nodD2-nolR mutant. The sentence (L401-402) ‘This result suggests that HH103 Nops prevent nodulation on L. japonicus but not on P. vulgaris’ should be revised accordingly.

Thank you very much for this interesting suggestion. In fact, we are currently constructing a set of HH103 double mutants in these 4 regulators (ttsI, syrM, nodD2, nolR) and we plan to analyze their symbiotic behavior in L. japonicus and P, vulgaris in comparison with HH103 and with the single mutants. We have modified the sentence you suggested.

Second, all the three mutants nodD2nolR and syrM are able to form nodules on P. vulgaris however to a lower level than R. tropici CIAT899. It would be interesting to see whether the double nodD2-nolR or triple nodD2-nolR-syrM mutant would form more nodules.

As mentioned above, this line of work is on the way.

Third, the three mutants nodD2nolR and syrM were shown to fix low amount of nitrogen on P. vulgaris. I suggest ARA values (L284, L292, L293) should be calculated per nodule so as to known whether the defect in nitrogen fixation is due to the lower number of nodules or to both a lower number of nodules and a lower nitrogenase activity per nodule.

We have followed your suggestion. As you can see, the defect in nitrogen fixation is due to both a lower number of nodules and a lower nitrogenase activity per nodule.

Finally, as mentioned above, a quadruple mutant nodD2-nolR-syrM-tssI would be interesting to construct to see if these mutations could allow HH103 to form more nodules and fix higher amount of nitrogen on P. vulgaris.

As mentioned before, we are currently constructing double mutants, but your suggestion is very interesting. Although this might not be an easy task, we will do our best.

Other comments

L92: ‘Although the SyrM and NodD2 proteins activate the expression 91 of a few genes’ contradicts the publication reference 18 : ‘Sinorhizobium fredii HH103 syrM inactivation affects the expression of a large number of genes”.

We have corrected the sentence in order to make clear that we were referring to the effect of these genes on the nod regulon.

L113: ‘Identification of Nod Factors and biological activity assays’ it is not clear how many times independently this analysis was done. Please indicate the number of independent measurements (biological replicates). This should have been done at least 3 times.

Yes, this experiment was performed three independent times, with at least 6 plantlets per treatment and assay. This has been clarified in Material and Methods.

L127-129: ‘Bacterial isolation from surface-sterilized nodules for analysing the identity of bacteria occupying the nodules and ARA (acetylene reduction assay) for assessing nitrogenase activity of nodules were performed as previously described [38]’ the reference 38 does not mentioned any details. Please refer to the proper study with all methodological details.

Reviewer is right. We have replaced the old reference 38 by Buendía-Clavería et al (1986). This new reference 38 provides all methodological details related to ARA and bacterial isolation from nodules.

L140, L406: ‘(Vinardell et al. 2004Fitas; Acosta-Jurado et al. 2019)’ references need to be formatted.

Corrected.

L133-134: ‘The nodD1 gene of S. fredii HH103, but not nodD2, restores Nod factors production and nodulation ability in a R. tropici CIAT 899 nodD1 nodD2 mutant’ this title should be changed since restoration of Nod Factor production was not tested for the complemented strain with NodD2.

Reviewer is right. We have change the title accordingly.

L141-142: ‘In order to investigate whether the nodD1 or nodD2 genes from S. fredii HH103 may restore Nod factors production and, consequently, nodulation ability in a R. tropici CIAT899 ΔnodD1 ΔnodD2 mutant’, for the same reason this sentence should be replaced by ‘In order to investigate whether the nodD1 or nodD2 genes from S. fredii HH103 may restore nodulation ability in a R. tropici CIAT899 ΔnodD1 ΔnodD2 mutant’.

Reviewer is right. We have change the sentence accordingly.

L177-180 :’ Surprisingly, a set of eighteen Nod factors not previously described for CIAT 899 were produced by the ΔnodD1D2 (pMUS296) strain, which points to a kind of NodD-dependent plasticity when synthesizing Nod factors in R. tropici CIAT 899 (Figure 2 and Dataset S1).’ This conclusion can be drawn only if the experiment was sufficiently repeated (three times independently).

Reviewer is right. Although we have performed this experiment two independent times and with two replicates each, further research is required for drawing this conclusion. We have removed this part and, as required by reviewer one, added some comments about the Nod factors produced by CIAT 899 and CIAT899 ΔnodD1D2 (pMUS296). Please, see lines 186-192.

L279: ‘The ttsI derivative of HH103, as the wild-type strain, only induced pseudonodules and non-colonised white nodules’, the bacterial colonization of nodules induced by the ttsI mutant is not shown in this paper. ‘non-colonised’ should be removed in this sentence unless cytological analysis of nodule sections have been done (or the number of bacteria/nodule determined) and shown in a figure.

This part of the text has been arranged accordingly to reviewer suggestion.

L281-283: ‘The HH103 281 syrM derivative was able to induce the formation of both non-colonised and colonised nodules’. This sentence should be removed unless the colonization data are shown in the paper (through cytological observations of nodule

sections or determination of the number of bacteria per nodule).

This part of the text has been arranged accordingly to reviewer suggestion.

L329: ‘acts a repressor’ should be ‘acts as a repressor’.

Corrected.

We hope that the new version of the manuscript will be acceptable for publication in International Journal of Molecular Sciences. In addition to thanking to you for your work as Section Managing Editor, we would like to thank again the two Reviewers for their work on this manuscript.

Best regards and Happy New Year,

Dr. José Mª Vinardell González

Full Professor

Department of Microbiology,

Faculty of Biology, University of Seville

Avda. Reina Mercedes 6, 41012-Sevilla (Spain)

+34 954554330

[email protected]

Reviewer 2 Report

In this paper, Fuentes-Romero and collaborators have analyzed the role of 5 symbiotic regulators (NodD1, NodD2, NolR, SyrM and TtsI) of S. fredii HH103, the natural symbiont of soybean but unable to nodulate efficiently L. japonicus and P. vulgaris. In previous works, authors showed that NodD1 is the main activator of nod genes in S. fredii, while NodD2, NolR and SyrM mainly act as repressors of the expression of nod genes and TtsI is a positive regulator of the T3SS of S. fredii. Moreover they previsouly showed that inactivation of the secondary regulators nodD2, syrM, nolR and ttsI allowed S. fredii to nodulate and infect L. janonicus. In this study, Fuentes-Romero and collaborators analyzed the role of these regulators on the symbiosis with P. vulgaris, another non-host plant of S. fredii, to see whether they play a negative role in this interaction like on L. japonicus.

Authors conducted this study in two ways. First, they analyzed the role of NodD1 (activator of nod genes) and nodD2 (repressor of nod genes) on the capacity to provide the nodulation ability of a double nodD1nodD2 mutant of R. tropici CIAT 899 a nitrogen-fixing symbiont of P. vulgaris. Introduction of nodD1, but not nodD2, on a plasmid allowed the nodulation of this strain on P. vulgaris (as well as on L. japonicus and L. Burtii) although with less efficiently than the wild-type strain, meaning that the S. fredii HH103 NodD1 is able to activate the nodulation genes of R. tropici CIAT899. Whether this lesser efficiency is due to the production of a different set of Nod Factors needs to be confirmed (see my comment below).

Besides, they analyze the role of the other regulators TtsI, NodD2, NolR and SyrM in S. fredii by constructing individual mutants of these genes in S. fredii HH103 and testing their nodulation and nitrogen fixation capacities on P. vulgaris. Authors found that a single tssI mutant does not allow nodulation on P. vulgaris while single mutants in nodD2, nolR or syrM partially restored nodulation on P. vulgaris. nodD2 and nolR mutants also partially restore the nitrogen fixation activity, indicating that these regulators negatively impact the interaction with P. vulgaris.

To strengthen the conclusions of the paper, I suggest some revisions.

First, I am wondering whether the effect of the ttsI inactivation is possible to observe in a strain that is not producing sufficient amounts of Nod Factors (inhibited by NodD2, NolR and SyrM). To reliably conclude that TtsI is not involved in host range restriction on P. vulgaris, the effect of the ttsI mutation should be tested in a nodD2 or nolR mutant or in a double nodD2-nolR mutant. The sentence (L401-402) ‘This result suggests that HH103 Nops prevent nodulation on L. japonicus but not on P. vulgaris’ should be revised accordingly.

Second, all the three mutants nodD2, nolR and syrM are able to form nodules on P. vulgaris however to a lower level than R. tropici CIAT899. It would be interesting to see whether the double nodD2-nolR or triple nodD2-nolR-syrM mutant would form more nodules.

Third, the three mutants nodD2, nolR and syrM were shown to fix low amount of nitrogen on P. vulgaris. I suggest ARA values (L284, L292, L293) should be calculated per nodule so as to known whether the defect in nitrogen fixation is due to the lower number of nodules or to both a lower number of nodules and a lower nitrogenase activity per nodule.

Finally, as mentioned above, a quadruple mutant nodD2-nolR-syrM-tssI would be interesting to construct to see if these mutations could allow HH103 to form more nodules and fix higher amount of nitrogen on P. vulgaris.

Other comments

L92: ‘Although the SyrM and NodD2 proteins activate the expression 91 of a few genes’ contradicts the publication reference 18 : ‘Sinorhizobium fredii HH103 syrM inactivation affects the expression of a large number of genes”.

L113: ‘Identification of Nod Factors and biological activity assays’ it is not clear how many times independently this analysis was done. Please indicate the number of independent measurements (biological replicates). This should have been done at least 3 times.

L127-129: ‘Bacterial isolation from surface-sterilized nodules for analysing the identity of bacteria occupying the nodules and ARA (acetylene reduction assay) for assessing nitrogenase activity of nodules were performed as previously described [38]’ the reference 38 does not mentioned any details. Please refer to the proper study with all methodological details.

L140, L406: ‘(Vinardell et al. 2004Fitas; Acosta-Jurado et al. 2019)’ references need to be formatted.

L133-134: ‘The nodD1 gene of S. fredii HH103, but not nodD2, restores Nod factors production and nodulation ability in a R. tropici CIAT 899 nodD1 nodD2 mutant’ this title should be changed since restoration of Nod Factor production was not tested for the complemented strain with NodD2.

L141-142: ‘In order to investigate whether the nodD1 or nodD2 genes from S. fredii HH103 may restore Nod factors production and, consequently, nodulation ability in a R. tropici CIAT899 ΔnodD1 ΔnodD2 mutant’, for the same reason this sentence should be replaced by ‘In order to investigate whether the nodD1 or nodD2 genes from S. fredii HH103 may restore nodulation ability in a R. tropici CIAT899 ΔnodD1 ΔnodD2 mutant’.

L177-180 :’ Surprisingly, a set of eighteen Nod factors not previously described for CIAT 899 were produced by the ΔnodD1D2 (pMUS296) strain, which points to a kind of NodD-dependent plasticity when synthesizing Nod factors in R. tropici CIAT 899 (Figure 2 and Dataset S1).’ This conclusion can be drawn only if the experiment was sufficiently repeated (three times independently).

L279: ‘The ttsI derivative of HH103, as the wild-type strain, only induced pseudonodules and non-colonised white nodules’, the bacterial colonization of nodules induced by the ttsI mutant is not shown in this paper. ‘non-colonised’ should be removed in this sentence unless cytological analysis of nodule sections have been done (or the number of bacteria/nodule determined) and shown in a figure.

L281-283: ‘The HH103 281 syrM derivative was able to induce the formation of both non-colonised and colonised nodules’. This sentence should be removed unless the colonization data are shown in the paper (through cytological observations of nodule sections or determination of the number of bacteria per nodule).

L329: ‘acts a repressor’ should be ‘acts as a repressor’.

Author Response

(The authors gave the same response as above.)

Round 2

Reviewer 2 Report

In the revised version of the manuscript entitled “The nodD1 gene of Sinorhizobium fredii HH103 restores nodulation capacity on bean in a Rhizobium tropici CIAT 899 nodD1/nodD2 mutant, but the secondary symbiotic regulators nolR, nodD2 or syrM prevent HH103 to nodulate with this legume”, Fuentes-Romero and collaborators have taken most reviewer’s comments into account. The manuscript can be published in the present form. I only have some minor revisions listed below.

L190 : « We investigated »

L361 : “However it is, it is”

L438-442 are identical to L520-523, one copy should be removed or rephrased.

Author Response

Microorganisms-1532292, Cover letter of the revised version of the manuscript

Dear Ms. Lily Liang,

We have sent our revised version of manuscript ID microorganisms-1532292 (“The nodD1 gene of Sinorhizobium fredii HH103 restores nodulation capacity on bean in a Rhizobium tropici CIAT 899 nodD1 nodD2 mutant, but the secondary symbiotic regulators nolR, nodD2 or syrM prevent HH103 to nodulate with this legume”).

We would like to sincerely thank again the two reviewers for their really useful comments and suggestions and for carrying out their work so quickly. In our opinion, they have contributed to improve the quality of the manuscript.

These are our answers to the different questions posed by reviewer 2 in 2nd round.

In the revised version of the manuscript entitled “The nodD1 gene of Sinorhizobium fredii HH103 restores nodulation capacity on bean in a Rhizobium tropici CIAT 899 nodD1/nodD2 mutant, but the secondary symbiotic regulators nolR, nodD2 or syrM prevent HH103 to nodulate with this legume”, Fuentes-Romero and collaborators have taken most reviewer’s comments into account. The manuscript can be published in the present form. I only have some minor revisions listed below.

L190 : « We investigated »

Corrected

L361 : “However it is, it is”

Corrected

L438-442 are identical to L520-523, one copy should be removed or rephrased.

Reviewer is right. We have removed lines 438-422.

We hope that the new version of the manuscript will be acceptable for publication in International Journal of Molecular Sciences. In addition to thanking to you for your work as Section Managing Editor, we would like to thank again the two Reviewers for their work on this manuscript.

Best regards and Happy New Year,

Dr. José Mª Vinardell González

Full Professor

Department of Microbiology,

Faculty of Biology, University of Seville

Avda. Reina Mercedes 6, 41012-Sevilla (Spain)

+34 954554330

[email protected]